# Continual deep learning by Functional Regularisation of Memorable Past

## Abstract

Continually learning new skills without forgetting old ones is an important quality for an intelligent system, yet most deep learning methods suffer from catastrophic forgetting of the past. Recent works have addressed this by regularising the network weights, but it is challenging to identify weights crucial to avoid forgetting. A better approach is to directly regularise the network outputs at past inputs, e.g., by using Gaussian processes (GPs), but this is usually computationally challenging. In this paper, we propose a scalable *functional-regularisation* approach where we regularise only over a few *memorable past* examples that are crucial to avoid forgetting. Our key idea is to use a GP formulation of deep networks, enabling us to both identify the memorable past and regularise over them. Our method achieves state-of-the-art performance on standard benchmarks and opens a new direction for life-long learning where regularisation methods are naturally combined with memory-based methods.

## 1 Introduction

The ability to quickly adapt to changing environments is an important quality of intelligent systems. For such quick adaptation, it is important to be able to identify, memorise, and recall useful past experiences when acquiring new ones. Unfortunately, standard deep-learning methods are not good at maintaining previously acquired skills, and can quickly forget them when learning new skills (Kirkpatrick et al., 2017). Such catastrophic forgetting presents a big challenge when deploying deep-learning methods for applications, such as robotics, where new tasks can appear during the training, and data from the previous tasks might be unavailable for retraining.

In recent years, many methods have been proposed to address catastrophic forgetting in deep learning. One of the most popular approaches is to regularise the network weights to keep them close to the weights obtained for the previous tasks/data (Kirkpatrick et al., 2017; Nguyen et al., 2018; Zenke et al., 2017; Ebrahimi et al., 2019; Serra et al., 2018). This is challenging due to the difficulty in identifying the weights that are relevant to past tasks. The exact values of the weights in fact do not matter directly, but rather the network output (Benjamin et al., 2018). Figuring out which weight affects the output is therefore usually difficult. Typically, the Fisher information matrix or covariance matrices over weights are used (Kirkpatrick et al., 2017; Nguyen et al., 2018), but they only partially address the issue.

A better approach is to directly regularise the network outputs, also referred to as *functional-regularisation*, requiring a memory of past examples (Benjamin et al., 2018; Lopez-Paz & Ranzato, 2017; Rebuffi et al., 2017). However, such methods still lack a mechanism to automatically weight more relevant past memory in the context of the new task, and also do not take uncertainty of the output into account. Methods based on Gaussian processes do this automatically (Titsias et al., 2019), but require optimisation over inducing points and specification of a good kernel, both of which are difficult tasks. In summary, existing methods fall short in building scalable functional-regularisation methods for continual learning.

In this paper, we propose a new functional-regularisation method where we regularise over a few *memorable past* examples (see Fig. 1). Our approach builds upon a recent method of Khan et al. (2019) that expresses deep networks as Gaussian processes (GPs). We show that the GP formulation not only enables the identification of examples crucial to avoid forgetting, but also computes uncertainty over the network output to appropriately weight the past examples in the light of new

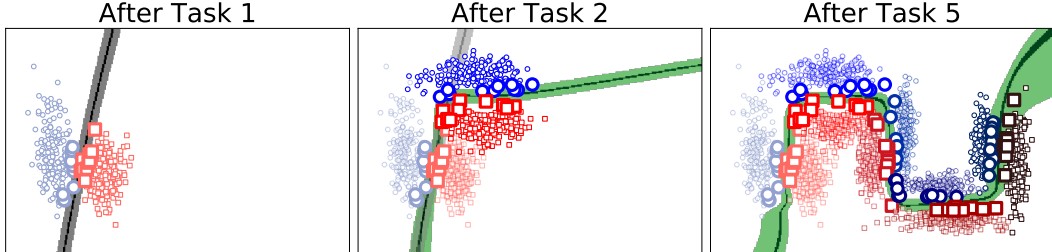

| After Task 1 | After Task 2 | After Task 5 |
|---|---|---|

Figure 1: This figure illustrates our method. Leftmost figure shows the result of training on task 1. Examples corresponding to memorable-past, shown with big markers, are chosen using a GP formulation of the neural network. These points usually are the ones that support the decision boundary. Middle figure shows the result after task 2 where new network functions are regularised at memorable-past examples to give the same prediction as the previous ones. The resulting green decision boundary classifies both task 1 and 2 well. The rightmost figure shows the result along with memorable-past of each task where the performance over the past tasks is maintained.

ones. Motivated by the GP perspective, we propose a new loss function for function-regularisation with deep networks. A Laplace approximation of this objective enables scalable training. Our work in this paper focusses on avoiding forgetting, but it opens a new direction for life-long learning methods where regularisation methods are naturally combined with memory-based methods.

**Other related works.** Broadly, existing work on continual learning can be split into three approaches: inference based, memory/rehearsal based, and model based. Inference based approaches have mostly focused on weight regularisation, with some recent efforts on functional regularisation. Our work falls in the latter category. Memory based approaches either maintain a memory of past data examples (Rebuffi et al., 2017) or train generative models on previous tasks to rehearse pseudo-inputs (Shin et al., 2017). An advantage of our method compared to previous ones is that building memory does not require solving an optimisation problem: the computation simply involves a forward-pass through the network followed by sorting (see Section 3.2).

Similarly to our work, there have also been some efforts in combining the different flavours of approaching continual learning, e.g., VCL plus coresets (Nguyen et al., 2018) and Gradient-Episodic Memory (Lopez-Paz & Ranzato, 2017; Chaudhry et al., 2018). Benjamin et al. (2018) have proposed a similar combination for functional regularisation. In these approaches, two separate methods are usually used for regularisation and memory-building. Recent follow-up work (Aljundi et al., 2019; Chaudhry et al., 2019) has focussed on improving memory-building methods, but this separation has remained the case. In contrast, in our approach, both of these are done within the same GP framework by using the method of Khan et al. (2019).

Finally, model based approaches change the model architecture during training (Rusu et al., 2016), and this can be combined with other approaches (Schwarz et al., 2018). It is possible to use similar features in our GP based framework, which is an interesting future direction to be pursued.

## 2 CONTINUAL LEARNING WITH WEIGHT/FUNCTIONAL REGULARISATION

In deep learning, we minimise loss functions to estimate network weights. For example, in supervised multi-class classification problems, we are given a dataset $\mathcal{D}$ of $N$ input-output pairs with outputs $\mathbf{y}_i$, a vector of $K$ classes, and inputs $\mathbf{x}_i$, a vector of length $D$, and our goal is to minimise a loss which takes the following form: $\bar{\ell}(\mathbf{w}) + \delta R(\mathbf{w})$, where $\bar{\ell}(\mathbf{w}) := \frac{1}{N} \sum_{i=1}^{N} \ell(\mathbf{y}_i, \mathbf{f}_w(\mathbf{x}_i))$ with deep neural network $\mathbf{f}_w(\mathbf{x}) \in \mathbb{R}^K$ and its weights $\mathbf{w}$. $\ell(\mathbf{y}, \hat{\mathbf{y}})$ denotes a differentiable loss function between an output $\mathbf{y}$ and its prediction $\hat{\mathbf{y}}$, $R(\mathbf{w})$ is a regularisation function (usually an $L_2$-regulariser $R(\mathbf{w}) = \mathbf{w}^\top \mathbf{w}$) and $\delta > 0$ controls the regularisation strength. Standard deep-learning approaches rely on an unbiased stochastic-gradient of the loss $\bar{\ell}$, which usually requires access to all of the data examples for all classes (Bottou, 2010). It is this unbiased, minibatch setting where deep-learning excels and achieves state-of-the-art performance on many benchmark datasets.

In reality, we do not always have access to all the data at once, and it is not possible to obtain unbiased stochastic gradients. New classes may appear during training and old classes may never be seen again. For such settings, vanilla mini-batch stochastic-gradient methods lead to catastrophic forgetting of past information (Kirkpatrick et al., 2017). Our goal in this paper is to design methods that can avoid such catastrophic forgetting. We focus on a particular setting where the classification task is divided into several tasks, e.g., a task may consist of a classification problem over a subset of classes. We assume that the tasks arrive sequentially one after the other. Once the learning is over, we may never see that task again. Such continual-learning settings have been considered in previous works (Kirkpatrick et al., 2017; Nguyen et al., 2018; Zenke et al., 2017), and our goal is to avoid forgetting of old tasks in this setting.

Recent methods have proposed weight-regularisation as a way to combat catastrophic forgetting. The main idea is to keep the new network weights close to the old ones, e.g., when training a task $t$ while given network weights $\mathbf{w}_{t-1}$ trained on task $t - 1$, we can minimise the following loss: $\bar{\ell}_t(\mathbf{w}) + \delta(\mathbf{w} - \mathbf{w}_{t-1})^\top \mathbf{F}_t(\mathbf{w} - \mathbf{w}_{t-1})$, where $\bar{\ell}_t(\mathbf{w})$ is the loss defined over all data examples from task $t$ and $\mathbf{F}_t$ is a preconditioning matrix that favors the weights relevant to the past tasks more than the rest. The Elastic-Weight Consolidation (EWC) method (Kirkpatrick et al., 2017), for example, uses the Fisher information matrix as the pre-conditioner, while Ritter et al. (2018) use the Hessian of the loss, and VCL (Nguyen et al., 2018) essentially employs the precision matrix of the variational approximation to do the same. Such weight-space methods reduce forgetting but do not produce satisfactory results.

The challenge in using weight-regularisation lies in the fact that the exact values of the weights do not really matter due to parametric symmetries (Benjamin et al., 2018; Bishop, 2006). Since only the network outputs matter, an alternative approach is to directly regularise these. Benjamin et al. (2018) propose to use an $L_2$-regulariser over the function values on data examples from past tasks:

$$\min_w \bar{\ell}_t(\mathbf{w}) + \delta \sum_{s=1}^{t-1} \sum_{i \in \mathcal{M}_s} \| f_w(\mathbf{x}_i) - f_{w_{t-1}}(\mathbf{x}_i) \|_2^2, \tag{1}$$

where $\mathcal{M}_s$ is the set of small examples for task $s$ stored in the *working memory* (Lopez-Paz & Ranzato, 2017; Rebuffi et al., 2017). One major issue with this approach is that the $L_2$-regulariser weights all the data points equally. In reality, some examples in the memory are more important than others to learn a given task. In addition, the uncertainty of the prediction is also ignored. As we will show, working with distributions over functions allows us to address these two issues.

Gaussian processes (GPs), for example, enable automatic reweighting of old tasks in light of a new one, which happens by using the posterior covariance. Unfortunately, both scalability of GPs as well as the necessity to save many instances of the past makes them impractical. A recent approach by Titsias et al. (2019) attempts to address these issue by employing sparse GP methods with inducing points and using a neural network feature map. However, their approach has some difficulties. Their approach heavily depends on a proper choice of inducing points, which are normally obtained via an ad-hoc procedure. Additionally, they propose using the last layer of the neural network as kernel features, which is limiting as it does not use the whole network's weights in the kernel.

Ideally, we would want to perform functional regularisation with a deep learning optimiser while borrowing ideas from the GP methods. Our work in this paper takes a step in this direction. Our proposal is to use a GP formulation of neural networks to perform functional regularisation that still allows training with standard deep-learning methods. We also show that the GP view helps in selecting an informative set of memory points.

## 3 FUNCTIONAL-REGULARISATION OF MEMORABLE PAST (FROMP)

We will now describe our proposed method. The first step is to use the GP formulation of Khan et al. (2019) to compute GP-like posteriors over functions. Then, we propose a method to identify a set of memorable past examples for a task. Finally, we describe our functional-regularisation and discuss approximations used to build a scalable method.

## 3.1 FROM DEEP NETWORKS TO GAUSSIAN PROCESS POSTERIORS

Khan et al. (2019) propose an approach to convert deep networks into Gaussian processes. Their main result (see Theorem 1 in their paper) states that, at a local minimiser $\mathbf{w}_*$ of $\bar{\ell}(\mathbf{w}) + \frac{\delta}{2}\mathbf{w}^\top\mathbf{w}$, a Laplace approximation of the posterior over $\mathbf{w}$ is equivalent to the posterior distribution of a linear model. Specifically, they use the following scalable variant of the Laplace approximation:

$$p(\mathbf{w}|\mathcal{D}) \approx q(\mathbf{w}) := \mathcal{N}(\mathbf{w}|\boldsymbol{\mu}, \boldsymbol{\Sigma}), \text{ where } \boldsymbol{\mu} = \mathbf{w}_* \text{ and } \boldsymbol{\Sigma}^{-1} = \sum_{i=1}^{N} \mathbf{J}(\mathbf{x}_i)^\top \boldsymbol{\Lambda}(\mathbf{x}_i, \mathbf{y}_i)\mathbf{J}(\mathbf{x}_i) + \delta\mathbf{I}_P, \quad (2)$$

where $\mathbf{J}(\mathbf{x}_i) := \nabla_w \mathbf{f}_w(\mathbf{x}_i)^\top$ is a $K \times P$ Jacobian matrix ($P$ being the number of parameters), and $\boldsymbol{\Lambda}(\mathbf{x}, \mathbf{y}) := \nabla^2_{\mathrm{ff}}\ell(\mathbf{y}, \mathbf{f})$ is the $K \times K$ Hessian of the loss with $\mathbf{f} = \mathbf{f}_w(\mathbf{x})$, all evaluated at $\mathbf{w} = \mathbf{w}_*$. They show that $q(\mathbf{w})$ is equal to the posterior of the following linear model:

$$\tilde{\mathbf{y}} = \mathbf{J}(\mathbf{x})\mathbf{w} + \boldsymbol{\epsilon}, \quad \text{with } \boldsymbol{\epsilon} \sim \mathcal{N}(0, (\boldsymbol{\Lambda}(\mathbf{x}, \mathbf{y}))^{-1}) \text{ and } \mathbf{w} \sim \mathcal{N}(0, \delta^{-1}\mathbf{I}_P), \quad (3)$$

where the observations are defined as $\tilde{\mathbf{y}}_i := \mathbf{J}(\mathbf{x}_i)\mathbf{w}_* - (\boldsymbol{\Lambda}(\mathbf{x}_i, \mathbf{y}_i))^{-1} \mathbf{r}(\mathbf{x}_i, \mathbf{y}_i)$ with the residual $\mathbf{r}(\mathbf{x}, \mathbf{y}) := \nabla_{\mathbf{f}}\ell(\mathbf{y}, \mathbf{f})$. They also show that the predictive distribution of this linear model is equivalent to that of a GP regression model defined with a $K \times K$ neural tangent kernel (NTK) (Jacot et al., 2018):

$$\tilde{\mathbf{y}} = \mathbf{f}_{\mathrm{GP}}(\mathbf{x}) + \boldsymbol{\epsilon}, \quad \text{with } \boldsymbol{\epsilon} \sim \mathcal{N}(0, (\boldsymbol{\Lambda}(\mathbf{x}, \mathbf{y}))^{-1}) \text{ and } \mathbf{f}_{\mathrm{GP}}(\mathbf{x}) \sim \mathcal{GP}\left(0, \delta^{-1}\mathbf{J}(\mathbf{x})\mathbf{J}(\mathbf{x}')^\top\right). \quad (4)$$

The above result gives a posterior in the defined observation space of $\tilde{\mathbf{y}}$. In the case where we can find a map from $\tilde{\mathbf{y}}$ to $\mathbf{y}$, the above inference problem allows for reparameterisation into the original data space. Therefore, we will obtain a GP that serves as an approximation to the neural network posterior. Below, we demonstrate this for binary classification with sigmoid link function and Bernoulli likelihood (see Appendix A for details and extension to multi-class classification).

We parameterise the Bernoulli likelihood by $p(\mathbf{x}) := \sigma(f_{w_*}(\mathbf{x}))$ where $\sigma$ is the sigmoid function (note that we now have scalars $\{f, y, ...\}$ as we are dealing with a single output). The residual and loss Hessian for this model are given by $r(\mathbf{x}, y) = p(\mathbf{x}) - y$ and $\Lambda(\mathbf{x}, y) = p(\mathbf{x})(1 - p(\mathbf{x}))$, respectively, where $y \in \{0, 1\}$. Because residuals are linear in $y$ and the Hessian is independent of $y$, we can substitute the definitions of $\tilde{y}$ and $r(\mathbf{x}, y)$ into Eq. 3, and rearrange for $y$,

$$y = \underbrace{p(\mathbf{x}) + \Lambda(\mathbf{x}, y)\mathbf{J}(\mathbf{x})(\mathbf{w} - \mathbf{w}_*)}_{:=f_{\mathrm{lin}}(\mathbf{x})} + \tau, \quad \text{with } \tau \sim \mathcal{N}(0, \Lambda(\mathbf{x}, y)) \text{ and } \mathbf{w} \sim \mathcal{N}(0, \delta^{-1}\mathbf{I}_P). \quad (5)$$

The Laplace approximation in Eq. 2 is the posterior of this model after observing data $\mathcal{D}$. We can equivalently write the posterior predictive as a GP (Rasmussen, 2003) with mean and covariance

$$m(\mathbf{x}) := \mathbb{E}_{q(w)}[f_{\mathrm{lin}}(\mathbf{x})] = p(\mathbf{x}, y) + \Lambda(\mathbf{x}, y)\mathbf{J}(\mathbf{x})(\boldsymbol{\mu} - \mathbf{w}_*) = p(\mathbf{x}), \quad (6)$$

$$k(\mathbf{x}, \mathbf{x}') := \mathbb{E}_{q(w)}[(f_{\mathrm{lin}}(\mathbf{x}) - m(\mathbf{x}))(f_{\mathrm{lin}}(\mathbf{x}') - m(\mathbf{x}'))]$$
$$= \Lambda(\mathbf{x}, y)\mathbf{J}(\mathbf{x})\mathbb{E}_{q(w)}[(\mathbf{w} - \mathbf{w}_*)(\mathbf{w} - \mathbf{w}_*)^\top]\mathbf{J}(\mathbf{x}')^\top\Lambda(\mathbf{x}', y')$$
$$= \Lambda(\mathbf{x}, y)\mathbf{J}(\mathbf{x})\boldsymbol{\Sigma}\mathbf{J}(\mathbf{x}')^\top\Lambda(\mathbf{x}', y'). \quad (7)$$

A GP defined with the above mean and covariance function can be viewed as an approximation to the posterior process over network outputs. Throughout the paper, we will denote the process obtained at a weight $\mathbf{w}_*$ by a distribution $q_{w_*}(\mathbf{f})$ where $\mathbf{f}$ is a vector of $f(\mathbf{x})$ evaluated at many different inputs $\mathbf{x}$. We will use this GP posterior predictive for functional regularisation.

## 3.2 MEMORABLE PAST

In the previous section, we derived a GP posterior approximation over the network outputs. Now, we propose a method to obtain a small set of examples that are crucial to avoid forgetting. We first note that the posterior GP mean in Eq. 4 corresponds to a kernel Ridge regression that requires the computation of $(\delta^{-1}\mathbf{K} + \boldsymbol{\Lambda}^{-1})^{-1}\tilde{\mathbf{y}}$ where $\boldsymbol{\Lambda}$ is a block-diagonal matrix containing all $\boldsymbol{\Lambda}_i := \boldsymbol{\Lambda}(\mathbf{x}_i, \mathbf{y}_i)$ and $\mathbf{K}$ is the NTK defined in Eq. 4. The predictions therefore strongly depend on the eigenvalues of the preconditioning matrix. Selection of important data examples therefore boils down to selecting

---

**Algorithm 1:** Functional Regularisation of Memorable Past (FROMP)

1: Initialise $\mathcal{M} \leftarrow \varnothing, \mathbf{s} \leftarrow \mathbf{0}, \mathbf{w} \leftarrow \mathbf{w}_0$.
2: **for** task $t = 1, 2, 3 \ldots, T$ **do**
3:    **for** previous task $s = 1, 2, ..., t-1$ **do**
4:       Precompute $\mathbf{m}_s, \mathbf{K}_s^{-1}$ using Eq. 6, 7
5:    **while** not converged **do**
6:       Stochastic gradients $\mathbf{g} \leftarrow \hat{\nabla}_w \ell(\mathbf{y}_i, \mathbf{f}_w(\mathbf{x}_i))$
7:       $\mathbf{g} \leftarrow \mathbf{g}+$ `fr_grad`$(\mathbf{w}, \mathbf{m}_{1:t-1}, \mathbf{K}_{1:t-1}^{-1})$
8:       Adam update for $\mathbf{w}$ using gradients $\mathbf{g}$
9:    $\mathbf{s} \leftarrow \mathbf{s} + \mathrm{diag}(\sum_{i \in \mathcal{D}_t} \mathbf{J}(\mathbf{x}_i)^\top \mathbf{\Lambda}(\mathbf{x}_i, \mathbf{y}_i) \mathbf{J}(\mathbf{x}_i))$
10:   $\mathbf{\Sigma} \leftarrow$ diagonal matrix with diagonal $1/(\mathbf{s} + \delta)$
11:   $\mathcal{M}_t \leftarrow$ `memorable_past` $(\mathcal{D}_t, \mathbf{w})$
12:   $\mathcal{M} \leftarrow \mathcal{M} \cup \mathcal{M}_t$

13: **function** `fr_grad`$(\mathbf{w}, \mathbf{m}_{1:t}, \mathbf{K}_{1:t}^{-1})$
14:   $\mathbf{g} \leftarrow \mathbf{0}$
15:   **for** task $s = 1, 2, ..., t$ **do**
16:     $\mathbf{f}_s \leftarrow$ vector of $\mathbf{f}_w(\mathbf{x}_i), \; \forall \mathbf{x}_i \in \mathcal{M}_s$
17:     $\mathbf{J}_s \leftarrow \nabla_w \mathbf{f}_s$
18:     $\mathbf{g} \leftarrow \mathbf{g} + \mathbf{J}_s \mathbf{K}_s^{-1} (\mathbf{f}_s - \mathbf{m}_s)$
19: **return** $\mathbf{g}$

20: **function** `memorable_past`$(\mathcal{D}_t, \mathbf{w})$
21:   Calculate $\mathbf{\Lambda}(\mathbf{x}_i, \mathbf{y}_i)$ for all $\mathbf{x}_i, \mathbf{y}_i$ in $\mathcal{D}_t$.
22: **return** $M$ examples with highest $\mathbf{\Lambda}(\mathbf{x}_i, \mathbf{y}_i)$.

---

Figure 2: A pseudo-code for our FROMP algorithm. The additional computations on top of Adam are in lines 4 and 7, where we add the contribution from the functional-regularisation term. After every task, we update the memorable past in Eq. 9-11 which involves a matrix inversion of size $M$, and a forward pass through the network to compute $\mathbf{\Lambda}(\mathbf{x}, \mathbf{y})$.

important columns of this matrix, such that the predictions remain unchanged. The theory of *leverage score sampling* (Alaoui & Mahoney, 2015; Bach, 2013) suggests picking the points proportional to the leverage score defined to be the diagonal of the following matrix: $\mathbf{K}(\mathbf{K} + \delta \mathbf{\Lambda}^{-1})^{-1}$. Typically, this matrix is difficult to compute and methods are employed to approximately obtain the leverage score (Alaoui & Mahoney, 2015).

For deep-learning applications too, exact computation of leverage score is very difficult. We instead propose a simple solution. Since the eigenvalues of the previous matrix heavily depend on $\delta \mathbf{\Lambda}_i^{-1}$, we can pick the data examples by simply sorting $\mathbf{\Lambda}_i$. For the inverse of $K + \delta \mathbf{\Lambda}^{-1}$ to have high eigenvalues, we should favour examples with smaller values of $\mathbf{\Lambda}_i^{-1}$. Therefore, we simply sort the $\mathbf{\Lambda}_i$ and pick the top $M$ examples as the most relevant ones. This simple solution is very effective for our particular problem because $\mathbf{\Lambda}_i$ are noise variances for the data examples, obtained by using an already trained network. These are second derivatives of the loss for data examples, and so also reflect the sensitivity of the decision boundaries if a particular data point is perturbed. Therefore, they tend to reflect the relevance of data examples. An example is shown in Figure 1 where we clearly see that our solution picks the examples lying close to decision boundary. Computation of $\mathbf{\Lambda}_i$ requires us to run the forward pass to get the $\ell(\mathbf{y}_i, \hat{\mathbf{y}}_i)$ and then compute its second derivative with respect to $\hat{\mathbf{y}}_i$, both of which are cheap operations. Throughout the paper, examples chosen by this method are referred to as the *memorable past* examples, and denoted $\mathcal{M}_t$ for task $t$.

### 3.3 FUNCTIONAL-REGULARISATION

So far, we described the construction of the GP posterior predictive over the function space, as well as the construction of a set of memorable-past examples. We are now ready to describe our objective function where we employ these two to perform functional regularisation.

Suppose that we are given network weights $\mathbf{w}_{t-1}$ that are obtained by training over data examples from task $t-1$. Our goal then is to train a network with weights $\mathbf{w}$ such that its performance on memorable past $\mathcal{M}_{1:t-1}$ is unchanged. We denote the vector of function outputs over these examples by $\mathbf{a}_{1:t-1}$. A straightforward idea is to directly optimise the weights $\mathbf{w}$ such that the predictions using $q_w(\mathbf{f}_t)$ are good on current tasks while the predictive distribution $q_w(\mathbf{a}_{1:t-1})$ is close to $q_{w_{t-1}}(\mathbf{a}_{1:t-1})$. Since the number of tasks can be very large, we choose to regularise each task separately, i.e., we will match $q_w(\mathbf{a}_s)$ with $q_{w_{t-1}}(\mathbf{a}_s)$ separately for all tasks $s < t$, in line with Titsias et al. (2019). These choices give us the following objective function with trade-off parameter $\tau$:

$$\min_w \quad \tau \mathbb{E}_{q_w(f_t)} \Big[ \sum_{i \in \mathcal{D}_t} \ell(y_i, f(\mathbf{x}_i)) \Big] + \sum_{s=1}^{t-1} \mathbb{D}_{KL}[q_w(\mathbf{a}_s) \, \| \, q_{w_{t-1}}(\mathbf{a}_s)]. \tag{8}$$

Table 1: Train accuracy of FROMP and batch-trained Adam (upper bound on performance) on variations of a toy 2D binary classification dataset, with mean and standard deviations over 10 runs (3 runs for Adam). FROMP performs well across variations. See Appendix D.2 for visualisations.

| Dataset variation | FROMP | Batch Adam |
|---|---|---|
| 10x less data (400 per task) | 99.9% ± 0.0 | 99.7% ± 0.2 |
| 10x more data (40000 per task) | 96.9% ± 3.0 | 99.7% ± 0.0 |
| Introduced 6th task | 97.8% ± 3.3 | 99.6% ± 0.1 |
| Increased std dev of each class distribution | 96.0% ± 2.4 | 96.9% ± 0.4 |
| 2 tasks have overlapping data | 90.1% ± 0.8 | 91.1% ± 0.3 |

A major issue with this objective is that computing gradients wrt. the posterior predictive $q_w$ will be computationally heavy as it involves gradients of the Jacobian and $\Lambda_i$, which involve higher order derivatives. During training on a task, we have no distribution on parameters (cf. Laplace approximation) and can treat it as fixed. Therefore, taking the gradient wrt. the parameter $\mathbf{w}$ of the above objective will only involve differentiating the mean function $\mathbf{m}(\mathbf{x}_i)$ formed by the neural network, and we have equivalence to the following objective:

$$\min_{w} \quad \tau \sum_{i \in \mathcal{D}_t} \ell(y_i, f_w(\mathbf{x}_i)) + \frac{1}{2} \sum_{s=1}^{t-1} (\mathbf{m}_{s,w} - \mathbf{m}_{s,w_{t-1}})^\top \mathbf{K}_{w_{t-1},s}^{-1} (\mathbf{m}_{s,w} - \mathbf{m}_{s,w_{t-1}}), \tag{9}$$

where $\mathbf{m}_{s,w_{t-1}}$ and $\mathbf{K}_{w_{t-1},s}$ are the vector and matrices containing the corresponding mean and covariance function of the network weights $\mathbf{w}_{t-1}$ evaluated at the memorable past for task $s$. Eqs 6 and 7 are used to calculate $\mathbf{m}_{s,w_{t-1}}$ and $\mathbf{K}_{w_{t-1},s}$ respectively. Essentially, this gives us something similar to Eq. 1 but instead of the $L_2$ distance, a kernel is used to improve weighting of examples. The final FROMP algorithm is summarised in Algorithm 1. Please see App. B for details on scaling to the multi-class setting.

The additional time complexity of this algorithm above vanilla Adam is $O(MPKt)$ per optimisation step, and $O([MPK + M^3]t + NPK)$ once per task. $O(NPK)$ is for updating the Laplace approximation $\Sigma$ at the end of training on a task, as the Jacobian for each input training point needs to be calculated (as in Eq. 2). This is similar to EWC (Kirkpatrick et al., 2017). Note that, as we use a diagonal approximation to $\Sigma$, inverting it is fast. $O(MPKt)$ is for calculating each of the $t$ matrices $\mathbf{K}_{w_{t-1},s}$, and $O(M^3t)$ is for inverting them. This is done once before training on each task (Eq. 7). Finally, during optimisation of the objective function, we need $O(MPK)$ for differentiating the mean function $\mathbf{m}(\mathbf{x}_i)$. All of these additional costs are small for small $M$, except for $O(NPK)$, which can be reduced by randomly sampling a subset of the task's data. Note that it is possible to be stochastic in sampling a subset of previous tasks' memory during every update, removing the time complexity dependency on $t$. Doing this gives the same overall complexity per step as in Adam (as $M$ is usually of the same order as the mini-batch size).

## 4 EXPERIMENTS

We run the proposed method FROMP on toy datasets, permuted MNIST and split MNIST (LeCun et al., 1998; Goodfellow et al., 2013), and a split version of CIFAR-10 and CIFAR-100 (Krizhevsky et al., 2009). We optimise the objective in Eq. 9 using Adam (Kingma & Ba, 2015) with parameter $\beta_1 = 0.99$ and further use gradient clipping to speed up training. To identify the individual benefits of the kernel in the loss (Eq. 9) as well as the selection of data points, we compare the following four methods: FROMP uses kernel and the proposed example selection, while FRORP instead uses randomly chosen examples. Further, we replace the kernel by an identity matrix leading to functional $L_2$ regularisation and call the corresponding methods with and without our example selection technique FROMP-$L_2$ and FRORP-$L_2$.

### 4.1 TOY DATASET

We first test FROMP on many variations of a toy 2D binary classification dataset like that in Figure 1. We want to test its performance when exposed to different datasets of varying difficulty. We use a

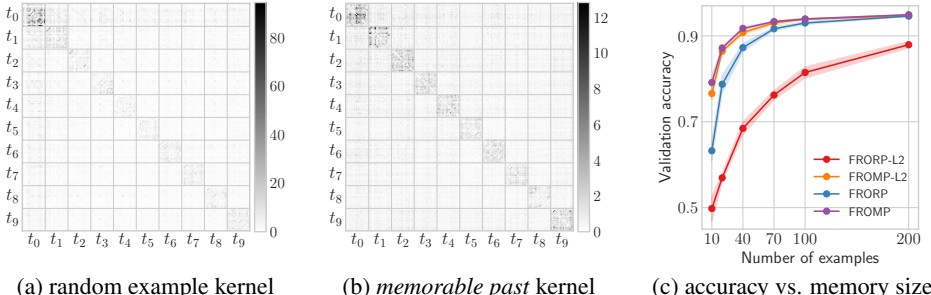

|(a) random example kernel|(b) *memorable past* kernel|(c) accuracy vs. memory size|

Figure 3: Permuted MNIST: added kernels across classes (with subtracted diagonal for visualisation purposes), and performance as a function of memory size (average accuracy after 10 tasks). Memorable past examples lead to a more uniform kernel structure that prevents weighting previously overfit examples too highly, e.g. task 1 in the random selection exhibits strong correlation and low variance. As we reduce the number of examples in memory, FROMP gracefully reduces accuracy.

2-hidden layer MLP (with 20 hidden units in each layer) for all experiments. Appendix D.3 details hyperparameter selection. In Appendix D.1 we show the brittleness and inconsistent behaviour of weight-space regularisation methods. In contrast, FROMP performs extremely well across many variations, showing consistently good results (see Table 1 and Appendix D.2 for visualisations).

Table 2: The average validation accuracy on Permuted-MNIST (10 tasks) and Split-MNIST. "200p/t" denotes that 200 examples are selected for each task. We report mean and standard deviations over 5 runs, and use results from Nguyen et al. (2018) for baselines. FROMP is state-of-the-art. Additionally, as we reduce the number of points (Figure 3), FROMP gracefully reduces accuracy, due to the clever choice of memorable past and the use of kernels in the functional regularisation.

| Method | Permuted MNIST | Split MNIST |
|---|---|---|
| DLP (Smola et al., 2003) | 82% | 61.2% |
| EWC (Kirkpatrick et al., 2017) | 84% | 63.1% |
| SI (Zenke et al., 2017) | 86% | 98.9% |
| Improved VCL (Swaroop et al., 2019) | 93% ± 1 | 98.4% ± 0.4 |
| + random Coreset | **94.6**% ± 0.3 (200 p/t) | 98.2% ± 0.4 (40 p/t) |
| FRCL-RND (Titsias et al., 2019) | 94.2% ± 0.1 (200 p/t) | 96.7% ± 1.0 (40 p/t) |
| FRCL-TR (Titsias et al., 2019) | 94.3% ± 0.1 (200 p/t) | 97.4% ± 0.6 (40 p/t) |
| FRORP-$L_2$ | 87.9% ± 0.7 (200 p/t) | 98.5% ± 0.2 (40 p/t) |
| FROMP-$L_2$ | 94.6% ± 0.1 (200 p/t) | 98.7% ± 0.1 (40 p/t) |
| FRORP | 94.6% ± 0.1 (200 p/t) | **99.0**% ± 0.1 (40 p/t) |
| FROMP | **94.9**% ± 0.1 (200 p/t) | **99.0**% ± 0.1 (40 p/t) |

## 4.2 PERMUTED AND SPLIT MNIST

Permuted MNIST consists of a series of tasks where each task is a fixed permutation of pixels to the entire labelled MNIST dataset. Like in previous work (Nguyen et al., 2018; Kirkpatrick et al., 2017; Zenke et al., 2017; Titsias et al., 2019), we implement a fully connected single-head network with two hidden layers, and report performance after 10 tasks. Each hidden layer consists of 100 hidden units and ReLU activation functions. We set the learning rate to 0.001, batch size to 128, and learn each task for 10 epochs. We use $\tau = 1$ for both FROMP and FROMP-$L_2$ (see Eq. 9).

The Split MNIST experiment was introduced by Zenke et al. (2017) and consists of five binary classification tasks built from MNIST: 0/1, 2/3, 4/5, 6/7, and 8/9. We use a fully connected multi-head neural network with two hidden layers, each with 256 hidden units and ReLU activation functions. Following the settings of previous work, we select 40 memorable points per task. The learning rate is set to 0.0001, batch size to 128, and we learn each task for 15 epochs. We find optimal parameters $\tau = 1$ and $\tau = 10$ for FROMP and FROMP-$L_2$ respectively.

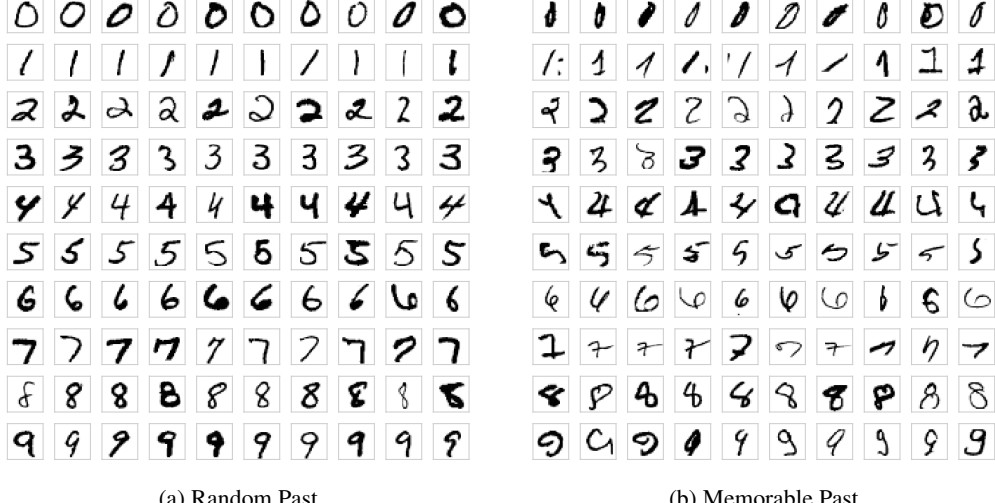

(a) Random Past                                                     (b) Memorable Past

Figure 4: Split MNIST: randomly selected samples and *memorable past* in comparison. 10 samples per class and method. In line with the toy example in Fig. 1, memorable past examples appear to be closer to the decision boundary of a classifier: many samples of the memorable past are possibly hard to distinguish from other classes.

We report the final average accuracy for both benchmarks across all the tasks in Table 2. The proposed method achieves better performance than the weight-space methods EWC and VCL, as well as compared to the function-space method FRCL that is based on a GP formulation. Further, the benchmarks show superior performance of both the approach to select important examples (Sec. 3.2) and the functional regularisation using the kernel (Sec. 3.3). Memorable examples improve performance of the naive and efficient FRORP-$L_2$ method by more than 6% on permuted MNIST and by 0.2% on the split MNIST. Standard deviation is also reduced in both cases. FROMP does not profit much from memorable examples compared to FRORP, probably because the performance is already close to the maximum achievable. Furthermore, Fig. 3c shows that our selection method greatly reduces the number of memorable points required: the $L_2$ algorithm with random points requires over 100 points to match the performance when using 20 carefully selected points, and 200 points to match performance with 40, respectively. When combined with kernel-based functional regularisation, we obtain the best-performing method, particularly when memory size is small. In line with the toy example in Fig. 1, Fig. 4 illustrates the memorable past examples compared to randomly selected examples. The memorable past examples appear to be more special cases, and may therefore lie closer to the decision boundary.

Figs. 3a and 3b show the summed kernels across all the classes in permuted MNIST for a random and memorable set of points. For visualisation purposes, the diagonal is suppressed. Note that random points lead to less uniform weighting in the kernel, making it even more important in functional regularisation, leading to better performance (FRORP vs FROMP). All memorable points are important and the kernel is more uniform. In Fig. 3a, the kernel's weighting of task 1 is very different from other tasks, leading to different magnitudes in the functional regularisation among tasks and therefore to eventual forgetting. The kernel further tells us that the tasks are correlated, as expected.

## 4.3 SPLIT CIFAR

Split CIFAR is far more complex than the MNIST experiments and consists of 6 tasks. The first task is the full CIFAR-10 dataset, followed by 5 tasks, each consisting of 10 consecutive classes from CIFAR-100. We follow the SI paper (Zenke et al., 2017) for our model architecture, using a multi-head CNN with 4 convolutional layers, followed by 2 dense layers with dropout. We use learning rate 0.0001 and batch size 256. All tasks are learned for 80 epochs and we use a memory of 10–200 examples. We use $\tau = 0.1$ and $\tau = 0.05$ for FROMP and FROMP-L2 respectively. In addition to continual learning baselines, we report the performance of networks trained from scratch

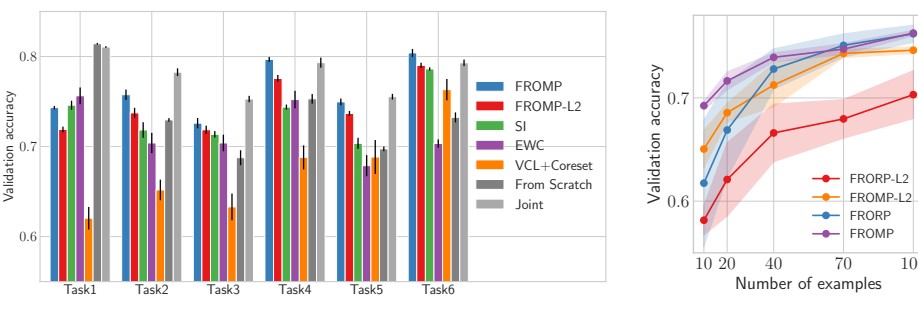

(a) final performance on individual tasks      (b) accuracy vs. memory size

Figure 5: Split CIFAR: performance for each task, and performance variation as function of memory size, after training on the final task. We run all methods 5 times and report the mean and standard error. For baselines, we train from scratch on each task and jointly on all tasks achieving $73.6\% \pm 0.4$ and $78.1\% \pm 0.3$, respectively. The left figure reports results for 200 memory examples per task. The final average validation accuracy of FROMP is $76.2\% \pm 0.4$, FROMP-$L_2$ is $74.6\% \pm 0.4$, SI is $73.5\% \pm 0.5$ (result from Zenke et al. (2017)), EWC is $71.6\% \pm 0.9$, VCL + random coreset is $67.4\% \pm 1.4$. FROMP outperforms all other methods, and is close to the performance of the jointly trained model, particularly in tasks 4-6. Additionally, as we reduce the memory size, FROMP still performs well, even with only 20 examples per task.

on each task. These cannot profit from forward/backward transfer. We also report the performance of a network jointly trained on all tasks. This is an upper bound on performance.

The experimental results in Fig. 5a show that FROMP outperforms other continual learning methods by a notable margin. The weight-space methods employed as baselines either cannot learn later tasks to a high accuracy (EWC, SI), or forget previous tasks (VCL). Interestingly, averaged over all tasks, FROMP also out-performs 'from scratch' training and achieves performance close to a jointly training on all tasks by a margin of less than 2%. Note that on tasks 4-6, FROMP achieves the same accuracy as the jointly trained model, showing no forgetting. We also calculate the backward transfer metric BWT from Lopez-Paz & Ranzato (2017). We find that FROMP has a score of $-2.6 \pm 0.9$, which is joint best with EWC's score of $-2.3 \pm 1.4$ (VCL+coresets has $-9.2 \pm 1.8$). Although EWC performs well in backward transfer, it does so at the cost of forward transfer. We calculate a forward transfer metric as the average of the improvement in accuracy on a new task over an independently trained model on that task (see App. C for more precise definitions of these metrics). FROMP achieves a forward transfer of $6.1 \pm 0.7$, whereas EWC has $0.17 \pm 0.9$ and VCL+coresets has $1.8 \pm 3.1$. Although we only focussed on preventing catastrophic forgetting, we find evidence of forward transfer, a key requirement in continual learning. Overall, given final average accuracy, backward transfer and forward transfer, FROMP clearly outperforms the other baselines.

In contrast to the rather simple MNIST benchmarks, both the benefit of selecting memorable points as well as using the kernel are clearly visible in Fig. 5b. If we only memorise a few examples, the performance gap due to using the kernel is around 4%. The selection of memorable points according to our metric leads to an increase in performance of around 7%. Applying both kernel and memorable point selection increases the performance by up to 11%. Additionally, standard deviation is reduced when using memorable points or the kernel. It is clear that both parts of the proposed algorithm are vital in achieving state-of-the-art performance on this benchmark.

## 5 DISCUSSION

We propose FROMP, a scalable function-regularisation approach for continual learning. FROMP uses a GP formulation of neural networks to select memorable past examples, regularising them using a kernel, and achieving state-of-the-art performance across benchmarks. This work enables a new way of combining regularisation methods and memory-based methods in continual learning. Future research could investigate other ways of selecting a memorable past (e.g. fixed memory size), more efficient ways of calculating kernel matrices, and the case where data does not arrive in tasks.

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

## A    DETAILS ON DEEP NETWORKS TO GAUSSIAN PROCESSES

In this section, we detail the map that is presented in Sec. 3.1 to go from the posterior of a deep neural network to a Gaussian process. The map for linear regression and logistic regression can be found in the appendix of Khan et al. (2019). We recap the map for logistic regression and, in addition, walk through the case of softmax regression here. Recall the linear model from Eq. 3:

$$\text{Model:} \quad \tilde{\mathbf{y}} = \mathbf{J}(\mathbf{x})\mathbf{w} + \boldsymbol{\epsilon}, \quad \text{with } \boldsymbol{\epsilon} \sim \mathcal{N}(0, (\mathbf{\Lambda}(\mathbf{x}, \mathbf{y}))^{-1}) \text{ and } \mathbf{w} \sim \mathcal{N}(0, \delta^{-1}\mathbf{I}_P) \tag{10}$$

$$\text{Data:} \quad \tilde{\mathbf{y}}_i := \mathbf{J}(\mathbf{x}_i)\mathbf{w}_* - \left(\mathbf{\Lambda}(\mathbf{x}_i, \mathbf{y}_i)\right)^{-1} \mathbf{r}(\mathbf{x}_i, \mathbf{y}_i). \tag{11}$$

Inference in this model yields the deep neural network posterior $q(\mathbf{w})$ in Eq. 2.

**Logistic regression:** as mentioned in Sec. 3.1, the parameter of the Bernoulli likelihood is given by applying the sigmoid link function to f($\mathbf{x}$) and we defined p($\mathbf{x}$) := $\sigma(\mathrm{f}_{w_*}(\mathbf{x}))$. Further, the derivative and Hessian of loss were particularised as r($\mathbf{x}, y$) = p($\mathbf{x}$) − y and $\Lambda(\mathbf{x}, y)$ = p($\mathbf{x}$)(1 − p($\mathbf{x}$)). We write $\varepsilon \sim \mathcal{N}(0, 1)$ for standard noise and plug the quantities model (Eq. 10):

$$\mathbf{J}(\mathbf{x})\mathbf{w}_* - \frac{\mathrm{p}(\mathbf{x}) - y}{\Lambda(\mathbf{x}, y)} = \mathbf{J}(\mathbf{x})\mathbf{w} + (\Lambda(\mathbf{x}, y))^{-\frac{1}{2}} \varepsilon_i, \tag{12}$$

which can be re-arranged to a new model that has an equivalent posterior, but where inference is done by observing original data points $\mathcal{D}$:

$$y = \mathrm{p}(\mathbf{x}) + \Lambda(\mathbf{x}, y)\mathbf{J}(\mathbf{x})(\mathbf{w} - \mathbf{w}_*) + \tau, \quad \text{with } \tau \sim \mathcal{N}(0, \Lambda(\mathbf{x}, y)) \text{ and } \mathbf{w} \sim \mathcal{N}(0, \delta^{-1}\mathbf{I}_P). \tag{13}$$

**Softmax regression** is a simple extension to the logistic regression case: we have a categorical likelihood with parameter $\mathbf{p}(\mathbf{x})$ := softmax($\mathbf{f}_{w_*}(\mathbf{x})$) and labels $\mathbf{y}_i$ that are standard one-hot encoded basis vectors, i.e. have a single 1 at some position and otherwise zeros. Then we have $\mathbf{r}(\mathbf{x}, \mathbf{y})$ = $\mathbf{p}(\mathbf{x}) - \mathbf{y}$ and $\mathbf{\Lambda}(\mathbf{x}, \mathbf{y})$ = diag($\mathbf{p}(\mathbf{x})$) − $\mathbf{p}(\mathbf{x})\mathbf{p}(\mathbf{x})^\top$. Unfortunately, $\mathbf{\Lambda}(\mathbf{x}, \mathbf{y})$ is of rank $K - 1$ and therefore we cannot uniquely invert it. The theory of Khan et al. (2019) therefore does not allow this because it requires $\mathbf{\Lambda}(\mathbf{x}, \mathbf{y}) > 0$. However, we can still write a model in line with Eq. 13 that we expect to have a posterior close to that of the deep neural network:

$$\mathbf{y} = \mathbf{p}(\mathbf{x}) + \mathbf{\Lambda}(\mathbf{x}, \mathbf{y})\mathbf{J}(\mathbf{x})(\mathbf{w} - \mathbf{w}_*) + \boldsymbol{\tau} \quad \text{with } \boldsymbol{\tau} \sim \mathcal{N}(0, \mathbf{\Lambda}(\mathbf{x}, \mathbf{y})) \text{ and } \mathbf{w} \sim \mathcal{N}(0, \delta^{-1}\mathbf{I}_P). \tag{14}$$

**Conversion to GP** can be achieved by taking the expectation and covariance of the linear model. To obtain the GP prior, we take the expectation of the linear model (10) over the prior $p(\mathbf{w})$. To obtain the posterior, we take the expectation over $q(\mathbf{w})$. Since we are interested in the posterior process, the latter is of interest and calculated for the logistic case in Sec. 3.1. The softmax regression case works equivalently but yields a $K \times K$ kernel, which can be represented as a symmetric tensor, or a $NK \times NK$ matrix over data set $\mathcal{D}$ with $N$ examples. To avoid growing complexity with more classes, we model the GPs for each class as being independent, as discussed in Appendix B.

## B    REDUCING COMPLEXITY IN THE MULTICLASS SETTING

For the softmax regression with $K$ classes, we build an individual GP for each class, in order to avoid growing complexity with more classes. We utilise $\mathrm{y}^{(k)}$ to denote the $k$-th item of $\mathbf{y}$ in Eq. (14). Then $\mathrm{y}^{(k)}$ will be:

$$\mathrm{y}^{(k)} = \mathrm{p}^{(k)} + \mathbf{\Lambda}(\mathbf{x}, \mathbf{y})^{(k)}\mathbf{J}(\mathbf{x})(\mathbf{w} - \mathbf{w}_*) + \tau^{(k)}. \tag{15}$$

Here $\mathbf{\Lambda}(\mathbf{x}, \mathbf{y})^{(k)}$ denotes the $k$-th row of the Hessian matrix. We transform the $K$ models to function space and treat them as independent. This allows us to split the KL-divergence in Eq. 8 into $K$ individual divergences, and we obtain the simplified final objective,

$$\min_{w} \quad \tau \sum_{i \in \mathcal{D}_t} \ell(y_i, f_w(\mathbf{x}_i)) + \frac{1}{2} \sum_{s=1}^{t-1} \sum_{k=1}^{K} (\mathbf{m}_{s,w}^{(k)} - \mathbf{m}_{s,w_{t-1}}^{(k)})^\top [\mathbf{K}_{w_{t-1},s}^{(k)}]^{-1} (\mathbf{m}_{s,w}^{(k)} - \mathbf{m}_{s,w_{t-1}}^{(k)}). \tag{16}$$

## C  FURTHER DETAIL ON CONTINUAL LEARNING METRICS REPORTED

We report a backward transfer metric and a forward transfer metric on split CIFAR. The backward transfer metric is exactly as defined in Lopez-Paz & Ranzato (2017). The forward transfer metric is a measure of how well the method uses previously seen knowledge to improve classification accuracy on newly seen tasks. Let there be a total of $T$ tasks. Let $R_{i,j}$ be the classification accuracy of the model on task $t_j$ after observing the last sample from task $t_i$. Let $R_i^{\text{ind}}$ be the classification accuracy of an independent model trained only on task $i$. Then,

$$\text{Backward Transfer, BWT} = \frac{1}{T-1} \sum_{i=1}^{T-1} R_{T,i} - R_{i,i},$$

$$\text{Forward Transfer} = \frac{1}{T-1} \sum_{i=2}^{T} R_{i,i} - R_i^{\text{ind}}.$$

## D  FURTHER TOY DATA EXPERIMENTS

This section provides further information and visualisations of toy 2D datasets, as well as hyperparameter settings for VCL.

### D.1  WEIGHT-SPACE REGULARISATION'S INCONSISTENT BEHAVIOUR

Table 3: Train accuracy of FROMP, VCL (no coresets), VCL+coresets and batch-trained Adam (an upper bound on performance) on a toy 2D binary classification dataset, with mean and standard deviations over 5 runs for VCL and batch Adam, and 10 runs for FROMP. 'VCL' is without coresets. VCL-RP and FRORP have the same (random) coreset selections. VCL-MP is provided with 'ideal' coreset points as chosen by an independent run of FROMP. VCL (no coreset) does very poorly, forgetting previous tasks. VCL+coresets is brittle with high standard deviations, while FROMP is stable.

| FROMP | FRORP | VCL-RP | VCL-MP | VCL | Batch Adam |
|---|---|---|---|---|---|
| 99.6% ± 0.2 | 98.5% ± 0.6 | 92% ± 10 | 85% ± 14 | 68% ± 8 | 99.70% ± 0.03 |

Table 3 summarises the performance (measured by train accuracy) of FROMP and VCL+coresets on a toy dataset similar to that in Figure 1. FROMP is very consistent, while VCL (with coresets) is extremely brittle: it can perform well sometimes (1 run out of 5), but usually does not (4 runs out of 5). This is regardless of coreset points chosen for VCL. Without coresets, VCL forgets many past tasks, with very low performance. Previous work (Farquhar & Gal, 2019) has argued that this is inevitable for weight-regularisation-only methods such as VCL and EWC.

We now 3 runs with different random seeds of VCL-MP from Table 3: the coreset is chosen from an independent run of FROMP, with datapoints all on the task boundary. This selection of coreset is intuitively better than a random coreset selection. Please note that the results we show here are not specific to coreset selection. Any coreset selection (whether random or otherwise) all show the same inconsistency when VCL is run on them. This behaviour is not specific to coreset choice.

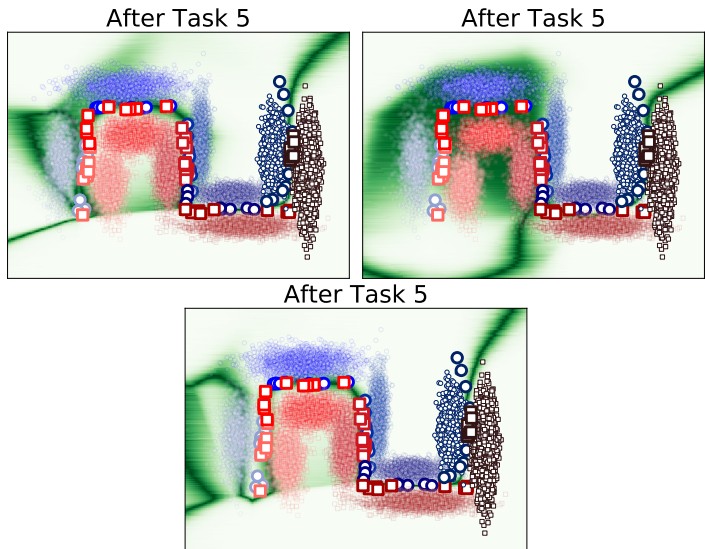

Figure 6: Three runs of VCL-MP on toy 2D data. These are the middle performing 3 runs out of 5 runs with different random seeds. VCL's inconsistent behaviour is clear.

## D.2 DATASET VARIATIONS

This section visualises the different dataset variations presented in Table 1. We pick the middle performing FROMP run (out of 5) and batch Adam run to show.

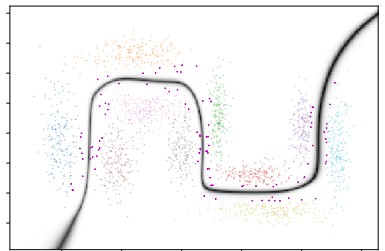 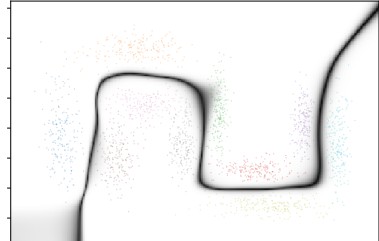

Figure 7: FROMP (middle performing of 5 runs) and batch Adam on a dataset 10x smaller (400 points per task).

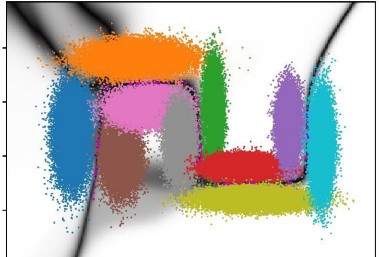 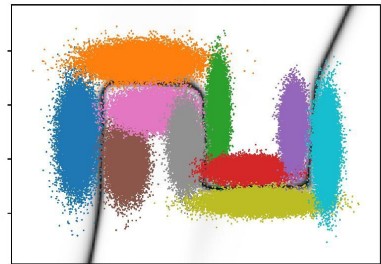

Figure 8: FROMP (middle performing of 5 runs), left, and batch Adam, right, on a dataset 10x larger (40,000 points per task).

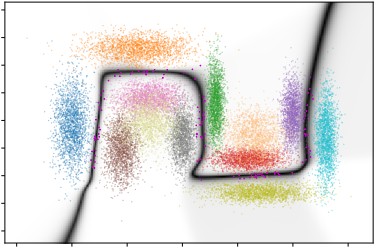 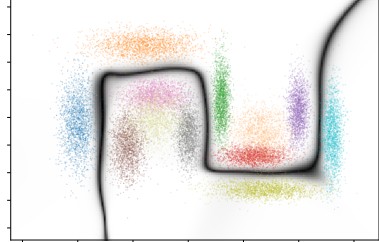

Figure 9: FROMP (middle performing of 5 runs), left, and batch Adam, right, on a dataset with a new, easy, 6th task.

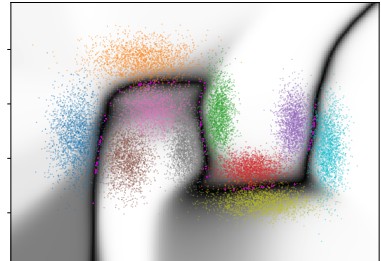 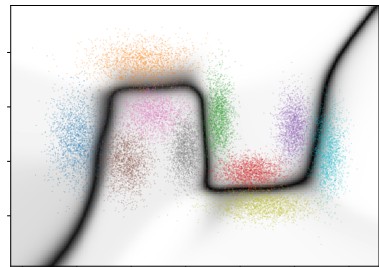

Figure 10: FROMP (middle performing of 5 runs), left, and batch Adam, right, on a dataset with increased standard deviations of each class' points, making classification tougher.

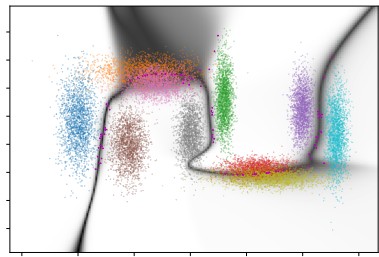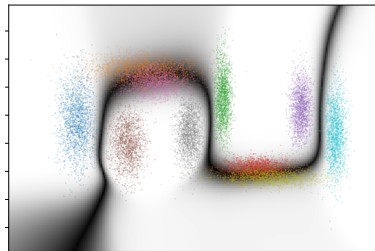

Figure 11: FROMP (middle performing of 5 runs), left, and batch Adam, right, on a dataset with 2 tasks having overlapping data, which is not separable.

### D.3 VCL AND FROMP HYPERPARAMETER SETTINGS FOR TOY DATASETS

**FROMP.** We optimised the number of epochs, Adam learning rate, and batch size. We optimised by running various settings for 5 runs and picking the settings with largest mean train accuracy on the toy dataset in Figure 1. We found the best settings were: number of epochs=50, batch size=20, learning rate=0.01. The hyperparameters were then fixed across all toy data experimental runs, including across dataset variations (number of epochs was appropriately scaled by 10 if dataset size was scaled by 10).

**VCL+coresets.** We optimised the number of epochs, the number of coreset epochs (because VCL+coresets trains on non-coreset data first, then on coreset data just before test-time: see Nguyen et al. (2018)), learning rate (we use Adam to optimise the means and standard deviations of each parameter), batch size, and prior variance. We optimised by running various settings for 5 runs and picking the settings with largest mean train accuracy. We found the best settings were: number of epochs=200, number of coreset epochs=200, a standard normal prior (variance=1), batch size=40, learning rate=0.01. VCL is slow to run (an order of magnitude longer) compared to all other methods (FROMP and batch Adam).

