# OpenReview forum: "Continual Deep Learning by Functional Regularisation of Memorable Past"
_ICLR.cc/2020/Conference — Reject_

### Official Review · AnonReviewer2 · 2019-10-23
**Official Blind Review #2**

**Rating:** 1

**Review:**

The paper proposed a new functional regularization method with gaussian process which has similar direction with recent two works (khan et al, titsias et al).
To perform functional regularization, they introduce small coreset which are selected from previous dataset instances, called memorable past. They select most memorable samples depends on eigenvalue. The model FROMP outperforms baselines and their ablations. However, the experiments are only performed on shallow networks, it is required to apply on much deeper networks, such as ResNet. Also, in the experiment results, I feel the performance of the FROMP largely depends on the number of the coreset, while 'important' selection just shows marginal effects even on split CIFAR.
FROMP show higher performance than FRORP with only a few of examples, but it isn't meaningful results that anyway the performances are too poor that are even worse than old baseline, EWC.

I have several wonderings on the paper.

- How about of training time on FROMP? I wonder if utilizing or selecting memorable pasts requires much time for training.

- Is there an analysis like figure 1 on real dataset, such as MNIST or CIFAR?





**Experience Assessment:**

I have published one or two papers in this area.

**Review Assessment: Checking Correctness Of Derivations And Theory:**

I carefully checked the derivations and theory.

**Review Assessment: Checking Correctness Of Experiments:**

I carefully checked the experiments.

**Review Assessment: Thoroughness In Paper Reading:**

I read the paper at least twice and used my best judgement in assessing the paper.

---

> ### Author Response · Authors · 2019-11-08
> **Response to AnonReviewer2**
>
> Thank you for your review. Our response is given below.
>
> Regarding the use of deeper networks: This paper develops a new method for continual learning. In line with previous literature in the area, we first evaluate the method on standard continual learning benchmarks established to evaluate new methods including EWC, SI, VCL and FRCL. In terms of scaling up, our current implementation requires Jacobian computation, which requires additional implementation to speed up computation. We hope to do this in the future.
>
> Regarding the dependance on the number of the coreset: We believe there is a misunderstanding about our experiments. Coresets are an important part of FROMP, and are the only way in which past information is propagated. Using very few coresets is not meaningful since there is very little to “remember” from the past. Comparing such small-size coreset cases to methods “without coresets” is not meaningful as these are complementary approaches, not direct competitors. As the coreset size gets very large, the selection strategy is not expected to matter. The purpose of Fig. 3c and 4b is to show that increasing coreset size improves results as expected, and using selected coresets rather than random is useful when the size of coreset is small. For example, selectively choosing a coreset of size 10 is about the same as randomly choosing 30 (on split CIFAR, Figure 4b). The ultimate number of coreset examples depends on the problem (e.g. data and network size). We are happy to discuss this further if this is unclear. Thanks!
>
> Training time: We will add a discussion in the paper. Our algorithm only adds a small computational overhead on top of Adam on a standard neural network. The additional complexity scales cubically in M, the coreset size. This is due to the inversion of the kernel in fr_grad. Another overhead is the computation of Jacobian which is order PKM, where K is the dimensionality of the output and P is the number of parameters. Both of these additional costs are small for small coreset sizes M.

---

### Official Review · AnonReviewer3 · 2019-10-23
**Official Blind Review #3**

**Rating:** 6

**Review:**

Summary
The paper proposes a method for continuous learning called Functional Regularization of Memorable Past (FROMP) which maintains the output distribution of models on memory samples. FROMP uses the Laplace approximation and Gaussian process with neural tangent kernel (NTK) to approximate the output distribution. According to the leverage score strategy, the sample to be stored is selected. The leverage score strategy tends to select the sample of highest variances.

Strengths
To some extent, I think the proposed method is novel, although there is a similar work named as Functional Regularisation for Continual Learning (FRCL). FROMP first uses NTK in Gaussian process for continual learning and proposes a new strategy of selecting memory samples.
The strategy of selecting samples to be stored is simple and effective.
The method achieves a good performance.
The paper is clearly written and easy to follow.

Weaknesses
It needs more experimental comparisons between FROMP and FRCL, like adding comparison results of FROMP and FRCL for Split-Cifar. Currently, this paper only shows the performance on Permuted MNIST and Split MNIST but those two benchmark are quite simple and also the improvement is limited.
The experimental section needs more detailed analysis. At least, in current version, it is not clear how many tasks in Permuted MNIST. The setting of hype-parameters for dropout are not provided.

Other comments
In this paper, for Split MNIST experiment with multi-head, it shows that the method of EWC achieves worse results than SI. However, in my experiment, the precision of EWC is at least larger than 97%. In theory, I think they should have the similar performance and at least the discrepancy of accuracy between them is not as big as shown in this paper. I expect authors could explain this point.

**Experience Assessment:**

I have published one or two papers in this area.

**Review Assessment: Checking Correctness Of Derivations And Theory:**

I assessed the sensibility of the derivations and theory.

**Review Assessment: Checking Correctness Of Experiments:**

I assessed the sensibility of the experiments.

**Review Assessment: Thoroughness In Paper Reading:**

I read the paper thoroughly.

---

> ### Author Response · Authors · 2019-11-08
> **Response to AnonReviewer3**
>
> Thanks for your comments about the strengths and weaknesses of our work. Our response is given below.
>
> We agree regarding the comparison with FRCL, but this is a very recent work and there is no available code. We will try to add this in the camera-ready but it will depend on the reproducibility of the FRCL paper (e.g. if they provide all the details necessary to reproduce results).
>
> We also agree on your comment about experimental details. We shall add them. For permuted MNIST, we used 10 tasks.
>
> Regarding your comment about EWC, could you provide a reference regarding this? We have reported the results from [1]. It is also possible that 97% is obtained with a much larger network than ours.
>
> [1] Nguyen, Cuong V et al. Variational continual learning. ICLR, 2018.

---

### Official Review · AnonReviewer1 · 2019-10-23
**Official Blind Review #1**

**Rating:** 1

**Review:**

Summary: The paper uses a Gaussian Processes framework previously introduced in [1] to identify the most important samples from the past for functional regularization. For evaluation authors report their average accuracy on Permuted MNIST, Split-MNIST, and CIFAR10-100 and achieve superior performance over EWC, DLP, SI, VCL-Coreset, and FRCL.

Pros:
(+): The paper is well-written, addressed the prior work quite well despite missing a few important work from the past (more on this later)
(+): The paper is well motivated

Cons that significantly affected my score and resulted in rejecting the paper are as follows:

1- lack of support for “scalability”:
Authors claim their method is scalable in several parts of the paper (abstract in line 7, Section 3 in the 1st paragraph, and Section 5 in Discussion). However, this claim is not supported in the experimental setting as the benchmark used are only toy datasets (Permuted MNIST, Split MNIST, and CIFAR10 followed by CIFAR100) where the maximum # of task considered is 10 and the maximum size of the datasets is 60K which is not convincing for ability to scale. There is also no time complexity provided.

2- Incremental novelty over the prior work (FRCL by Titsias et al 2019):
This baseline is the closest prior work to this work which according to the experiments shown in Table 2 are slightly outperformed by the proposed method. (for example for P-MNIST the gain is 0.6%+-0.1) where there is a lack of complete discussion on how the two methods are different. Particularly I suggest that the authors elaborate more on their claimed differences stated on page 4, paragraph 5 such as “tractability of the objective function only when we assume independence across tasks”. Do authors mean assuming clear task boundary between tasks? If so, have they considered a “no-task” or an "overlapping” task boundary in their experiment? Isn't it necessary to back up this if it is stated as a shortcoming of FRCL? Also, how are these methods differ in their computational expenses?

3- Lack of measuring forgetting:
This is the most important drawback in the experimental setting. Authors indicate on page 3 “Our goal in this paper is to design methods that can avoid such catastrophic forgetting.” and reiterate on this on other parts of the paper yet there is no forgetting evaluation to support this claim. Authors can simply report the initial performance of the model on each task so that readers can compare it with the reported accuracy after being done with all tasks. Having a method with high average accuracy does not necessarily mean it has minimum forgetting. You can use forgetting measurements such as Backward Transfer (BWT) introduced in [1] or forgetting ratio defined in [4] for this assessment.

4- Ambiguous claims about prior work:
(a) On page 1, paragraph 3, when authors mention that methods such as GEM or iCaRL use random selection to pick previous samples, I think the line of follow-up work on these methods should be mentioned as well that have explored different techniques for sample selection and have provided benchmark comparisons (ex. [2,3]). In fact it would be beneficial if authors could compare the samples selected by their method versus other sampling techniques.
(b) On page 1, paragraph 3, they mention some prior work such as GEM and iCaRL “do not take uncertainty of the output into account”. While it is true, there have been methods proposed that use uncertainty of the output for parameter regularization [5]. It appears to be a parallel work to this but it’s worth mentioning to prevent false claims.

5- Claim on the state of the art should be double-checked:
	Although the results shown for the experiments are superior to the provided baselines, there is an important baseline missing which has achieved higher performance than the reported ones. Also missed to be cited in the prior work list. Serra et al [4] proposed a method at ICML 2018 called HAT, which is a regularization technique with no memory usage that learns an attention mask over parameters and was shown to be very effective on small and long sequence of significantly different tasks. They do not use samples from previous task but yet achieved good average ACC as well as minimum forgetting ratio. Note that 5-Split MNIST is not reported in [4], but a recent work has reported HAT’s performance on this dataset (https://openreview.net/forum?id=HklUCCVKDB) that achieves 99.59%. I recommend authors provide comparison of their own on the given benchmarks with the original HAT’s implementation (https://github.com/joansj/hat) before claiming to be SoTA. In my opinion, it is not an issue if a novel method achieves a slightly lower performance to the sota because I think it still adds value and proposes a new direction. However, a false claim should not be stated.

Less major (only to help, and not necessarily part of my decision assessment):

1- Providing upper bound?
It is common to show an upper bound for any continual learning algorithm by showing joint training performance which is considered to be the maximum achievable performance. I also recommend showing the naive baseline of fine-tuning for the proposed method  which often can give insight to maximum forgetting ratio.

2- Forward transfer?
Regularization techniques combined with memory might have an ability to perform zero-shot transfer or so called FWT. I recommend authors provide such metric to further support their method.

3- Hyper parameter tuning?
It is also worth mentioning how the tuning process was performed. In continual learning we cannot assume that we have access to all tasks' data, hence authors might want to shed some light on this.

References:
[1] Khan, Mohammad Emtiyaz, et al. "Approximate Inference Turns Deep Networks into Gaussian Processes." arXiv preprint arXiv:1906.01930 (2019).

[2] Chaudhry, Arslan, et al. "Continual Learning with Tiny Episodic Memories." arXiv preprint arXiv:1902.10486 (2019). (https://arxiv.org/abs/1902.10486)

[3] Aljundi, Rahaf, et al. "Gradient based sample selection for online continual learning." arXiv preprint arXiv:1903.08671 (2019). (https://arxiv.org/abs/1903.08671)

[4] Serrà, J., Surís, D., Miron, M. & Karatzoglou, A.. (2018). Overcoming Catastrophic Forgetting with Hard Attention to the Task. Proceedings of the 35th International Conference on Machine Learning, in PMLR 80:4548-4557

[5] Ebrahimi, Sayna, et al. "Uncertainty-guided Continual Learning with Bayesian Neural Networks." arXiv preprint arXiv:1906.02425 (2019).


----------------------------------------------------------------------------------------------------------------------------------------------------------------------
----------------------------------------------------------------------------------------------------------------------------------------------------------------------
POST-REBUTTAL Response from R1:

Thank you for taking the time and replying to comments. Here are my responses to authors' replies:

[Authors' response:] 1. Scalability: Our algorithm only adds a small computational overhead on top of Adam on a standard neural network. This is what we mean by scalable. The additional complexity scales cubically in M, the coreset size. This is due to the inversion of the kernel in fr_grad. Another overhead is the computation of Jacobian which is order PKM, where K is the dimensionality of the output and P is the number of network parameters. Both of these additional costs are small for small coreset sizes M. We will add these details to make these points clear in the paper.

[Reviewer's response:] I still insist on the fact that simply explaining the overhead of a method is not a support for scalability claim versus showing the performance on a large scale dataset and comparing it with other CL methods that also have high scalability given the fact that authors only use MNIST and CIFAR datasets.

[Authors' response:] 4. Prior work: We discuss other works in Section 1 (“two separate methods are usually used for regularisation and memory-building”), and we will expand upon this sentence, going into more detail, and also referencing iCaRL and other works (including [3]). Note that our method of choosing a memorable past follows directly from the theory in Section 3.1, and is achieved with a single forward-pass through the trained network (as mentioned in the paper). Other techniques for sample selection do not integrate so naturally with the framework, and are not as straightforward to understand or implement either.

[Reviewer's response:] I disagree with authors on this because GEM, its faster version (A-GEM (Chaudhry et al. 2018)), and all other methods explored in the recent study which I mentioned in my review (Ref#2) use the single epoch protocol and are perfect match to be compared with this method but there is no memory-based baseline except for VCL with coreset and FRCL (only for MNIST variations) which makes it difficult to measure this method's capabilities (performance, memory size, and computational time) against methods which only require one epoch to be trained.

Authors have provided FWT for their method as 6% which is unbelievably large for this metric (see GEM paper) and hence does not make sense to me. Please double check whether you computed this value right.

While I accept the response for the remaining questions from authors but I am still concerned about the weak experiments and an issue brought up by R3 regarding lack of enough comparisons with FRCL on any other datasets besides split MNIST and P-MNIST. Also in  CIFAR experiment, what is the architecture used across the baselines? More importantly in results reported for VCL on CIFAR, it is not clear to me how authors obtained this results. Did they use a conv net? VCL was originally shown on MLPs only and it is one of the downside of this method that was never shown to be working in convolutional networks. Therefor, it is important to mention how they are obtained. This might explain the reason for the huge forgetting reported for VCL with coreset (−9.2 ± 1.8) as opposed to −2.3 ± 1.4 for EWC which is really strange as VCL even without coreset (on permuted mnist for example) is reported superior to EWC by a large margin (6%) in the original VCL paper. Overall I am concerned about the experimental setup and some of the reported results and hence intend to keep my score.

**Experience Assessment:**

I have published one or two papers in this area.

**Review Assessment: Checking Correctness Of Derivations And Theory:**

I assessed the sensibility of the derivations and theory.

**Review Assessment: Checking Correctness Of Experiments:**

I carefully checked the experiments.

**Review Assessment: Thoroughness In Paper Reading:**

I read the paper thoroughly.

---

> ### Author Response · Authors · 2019-11-08
> **Response to AnonReviewer1**
>
> Thank you for your long and useful review. We will first provide a short summary of our response, before going into more detail.
> - In terms of scalability, we test on standard small to medium size benchmarks, with complexity above Adam (on a standard neural network) dependent on M, the coreset size.
> - We will add more details comparing our method to FRCL [1], and provide a short summary below.
> - We will provide more metrics for measuring forgetting.
> - We will add a more detailed review of the literature on other coreset selection strategies, but unlike our strategy, these do not naturally fit within our framework.
> - We respectfully disagree with you on our claim about our method being state-of-the-art being false.
>
> 1. Scalability: Our algorithm only adds a small computational overhead on top of Adam on a standard neural network. This is what we mean by scalable. The additional complexity scales cubically in M, the coreset size. This is due to the inversion of the kernel in fr_grad. Another overhead is the computation of Jacobian which is order PKM, where K is the dimensionality of the output and P is the number of network parameters. Both of these additional costs are small for small coreset sizes M. We will add these details to make these points clear in the paper.
>
> 2. Comparison to FRCL [1]:
> (a) Thank you for raising this point. FRCL proposes using the last layer of the neural network as kernel features. This is limiting as it does not use the whole network’s weights, unlike what we do. A more important issue is with the difficulty of optimising inducing points; they are usually obtained by an ad-hoc procedure. In comparison, we provide a simple, effective way that is naturally consistent with our GP formulation. As per your suggestions, we will add a more detailed discussion explaining this.
> (b) There is a misunderstanding about our statement on “tractability of the objective function only when we assume independence across tasks”. This is not about the task boundaries. We mean that the GP used in FRCL defines separate kernels for each task, since otherwise the kernel is too big.
>
> 3. Measuring forgetting: Thank you for raising this point. We agree and will provide these. We are trying our best, but these may not be available by the end of rebuttal, in which case we will add them in the next version of the paper.
>
> 4. Prior work: We discuss other works in Section 1 (“two separate methods are usually used for regularisation and memory-building”), and we will expand upon this sentence, going into more detail, and also referencing iCaRL and other works (including [3]). Note that our method of choosing a memorable past follows directly from the theory in Section 3.1, and is achieved with a single forward-pass through the trained network (as mentioned in the paper). Other techniques for sample selection do not integrate so naturally with the framework, and are not as straightforward to understand or implement either.
>
> 5. Claim on state-of-the-art: We respectfully disagree with you on our claim being false. The 99.59% accuracy of HAT [2] on split MNIST is achieved with a much larger network. On the network we use, HAT achieves 91.6% on permuted MNIST, significantly lower than FROMP (94.9%), FRCL (94.3%) and VCL (93%). The openreview link you provided also uses a much larger network size (1200 units per hidden layer, as opposed to 256). We will add a reference to this work.
>
> [1] Titsias, Michalis K et al. Functional regularisation for continual learning using gaussian processes. arXiv preprint arXiv:1901.11356, 2019.
> [2] Serrà, J., Surís, D., Miron, M. & Karatzoglou, A.. (2018). Overcoming Catastrophic Forgetting with Hard Attention to the Task. Proceedings of the 35th International Conference on Machine Learning, in PMLR 80:4548-4557.
> [3] Ebrahimi, Sayna, et al. "Uncertainty-guided Continual Learning with Bayesian Neural Networks." arXiv preprint arXiv:1906.02425 (2019).

---

### Author Response · Authors · 2019-11-08
**Thanks to reviewers**

We would like to thank all the reviewers for their reviews, and the time they put into providing feedback. We will update the paper incorporating their feedback. We are in the process of obtaining some further metrics and visualisations as suggested by the reviewers, and will report them once we have them.

We will now address the points made by each reviewer in turn.

---

### Author Response · Authors · 2019-11-15
**Paper improvements**

We have update the paper in line with the reviewers' feedback.

Summary of changes:
- We added backward transfer and forward transfer metrics on split CIFAR for continual learning. They show how FROMP outperforms the baselines. We also added an upper bound of a model jointly trained on all tasks. FROMP performs close to this model, especially on tasks 4-6.
- We added a visualisation of memorable past vs random examples for split MNIST. The memorable past examples are harder to distinguish from other classes, in line with the toy example in Figure 1.
- We added a paragraph discussing the time complexity of our algorithm. It is small for small memorable past size M.
- We added a detailed discussion regarding FRCL.
- We added detailed hyperparameters for our experiments.
- We added more references and expanded upon some previous work (as suggested by AnonReviewer1).
- We cleaned up some notation and explanation in Section 3.3 and the Algorithm (and Appendix A). Please note that nothing technical has changed, and the overall algorithm is exactly the same.

Many thanks for your time.

---

### Decision · Program_Chairs · 2019-12-19

**Decision:**

Reject

**Comment:**

This work tackles the problem of catastrophic forgetting by using Gaussian processes to identify "memory samples" to regularize learning.

Although the approach seems promising and well-motivated, the reviewers ultimately felt that some claims, such as scalability, need stronger justifications. These justifications could come, for example, from further experiments, including ablation studies to gain insights. Making the paper more convincing in this way is particularly desirable since the directions taken by this paper largely overlap with recent literature (as argued by reviewers).